# Click Chemistry as a Tool for Cell Engineering and Drug Delivery

**DOI:** 10.3390/molecules24010172

**Published:** 2019-01-04

**Authors:** Yukiya Takayama, Kosuke Kusamori, Makiya Nishikawa

**Affiliations:** Laboratory of Biopharmaceutics, Faculty of Pharmaceutical Sciences, Tokyo University of Science, 2641 Yamazaki, Noda, Chiba 278-8510, Japan; 3a17702@ed.tus.ac.jp (Y.T.); makiya@rs.tus.ac.jp (M.N.)

**Keywords:** click chemistry, metabolic glycoengineering, cell surface modification, drug delivery, cancer therapy, cell tracking

## Abstract

Click chemistry has great potential for use in binding between nucleic acids, lipids, proteins, and other molecules, and has been used in many research fields because of its beneficial characteristics, including high yield, high specificity, and simplicity. The recent development of copper-free and less cytotoxic click chemistry reactions has allowed for the application of click chemistry to the field of medicine. Moreover, metabolic glycoengineering allows for the direct modification of living cells with substrates for click chemistry either in vitro or in vivo. As such, click chemistry has become a powerful tool for cell transplantation and drug delivery. In this review, we describe some applications of click chemistry for cell engineering in cell transplantation and for drug delivery in the diagnosis and treatment of diseases.

## 1. Introduction

Click chemistry is a term that was first proposed by Sharpless et al. in 2001. The characteristics of click chemistry include a high yield, a wide scope, less cytotoxic byproducts, a high stereospecificity, and a simple reaction [1]. Click chemistry reactions can occur under physiological conditions and the resulting chemical bonds are irreversible. Therefore, click chemistry is widely used for the modification of biomolecules, such as nucleic acids, lipids, and proteins with various compounds. Among the click chemistry reactions, the copper (I)-catalyzed azide-alkyne 1,3-dipolar cycloaddition (CuAAC) reaction has been used as a bioorthogonal reaction in the life science research fields (Scheme 1A) [2,3]. Moreover, the strain-promoted [3 + 2] azide-alkyne cycloaddition (SPAAC) reaction, which is a new type copper-free click chemistry developed by Bertozzi et al. in 2004, has brought about the successful application of click reactions to living cells without copper-induced cytotoxicity. They also reported that cyclooctyne (OCT) reacted with azide under physiological conditions without copper catalysis (Scheme 1B) [4,5]. However, the disadvantage of SPAAC reaction using OCT is that a long reaction time is required. The second-order rate constant of the reaction is 0.0024 M^−1^ s^−1^, which means that it takes over 120 min to sufficiently label azide-modified cells with OCT under physiological conditions [4]. To solve this problem, researchers have developed modified OCTs, including azadibenzocyclooctyne (ADIBO/DIBAC/DBCO) [6,7], biarylazacyclooctynone [8], bicyclo[6.1.0]nonyne (BCN) [9], dibenzocyclooctyne [10], and difluorinated cyclooctyne (DIFO) [11]. The second-order rates of these modified OCTs are about 24- to 400-fold greater than that of OCT and faster than that of the Staudinger reaction, a bioorthogonal reaction, under physiological conditions [5,12]. Furthermore, BCN and DBCO have a high solubility in water and a low affinity for serum proteins such as albumin. Therefore, copper-free click chemistry using modified OCTs is quicker, has a lower toxicity, and is widely recognized as a useful cell engineering method, in turn increasing the potential biological applications of click chemistry. In another study, Blackman et al. successfully developed the inverse electron demand Diels-Alder (iEDDA) reaction between the cycloaddition of s-tetrazine and trans-cyclooctene (TCO) derivatives, resulting in a faster copper-free click chemistry than SPAAC reactions (Scheme 1C) [13]. The second-order rate of 3,6-di-(2-pyridyl)-s-tetrazine with TCO is 2000 M^−1^ s^−1^ (in 9:1 methanol/water at 25 °C) and the reaction can take place in both water and cell culture media. Moreover, other researchers have developed bioorthogonal chemical reporters of the iEDDA reaction, including norbornene [14], cyclopropene [15,16], *N*-acylazetine [17], or vinylboronic acid [18], which react with tetrazines (Tz) under physiological conditions, and have demonstrated their usefulness for cell labeling with fluorophore and functional molecules. Importantly, these reagents hardly show toxicity to cells or animals at normal concentrations (we summarized in Table 1 and Table 2). Therefore, these rapid bioorthogonal iEDDA reactions are expected to be applied for cell engineering in biological field.

Metabolic engineering is a click chemistry tool that allows for the modification of living cells with chemical tags. Since biocomponents such as sugars, amino acids, or lipids are used and metabolized in living cells, using biomolecules with chemical tags can introduce chemical tags into proteins [19], glycans [20,21], and lipids [22] in living cells. Metabolic glycoengineering using sugar analogs is particularly useful for the introduction of SPAAC and iEDDA chemical substrates into living cells [21,23,24]. For example, monosaccharides with chemical tags can be incorporated into glycans in cells through a biosynthesis pathway to present chemical tags, including azide, alkene, and alkyne, on the surface of cells [21,25]. In metabolic glycoengineering, *N*-azidoacetyl-mannosamine (Ac_4_ManNAz) is widely used for engineering of the cell surface. Ac_4_ManNAz is metabolized into *N*-azidoacetyl neuraminic acid and is incorporated into glycans, such that azide groups are expressed in the glycans on the surface of living cells (Figure 1). Furthermore, Ac_4_ManNAz is not affected by the characteristics of culture cells (attachment, differentiation, migration, and mitochondria functions) with a lower toxicity in animal tissues [20,26,27,28,29]. Table 1 and Table 2 summarize the non-toxic dose ranges of the reagents reported in in vitro and in vivo studies using click chemistry and metabolic glycoengineering. Azide groups on cell surfaces after metabolic glycoengineering gradually disappear due to the hydrolysis of glycans by neuraminidase in cells after internalization [25,30]. However, it has been reported that azide groups were detected on the cell surface for at least 14 days [31], suggesting that metabolic glycoengineering is a suitable tool for cell labeling or functionalization through click chemistry. 

In this review, we describe the applications of click chemistry for cell engineering and drug delivery systems for the diagnosis and treatment of diseases.

## 2. Click Chemistry as a Tool for Cell Engineering in Cell Transplantation

Cell transplantation or cell-based therapy is a powerful therapeutic method for the treatment of various diseases including intractable diseases. Recently, clinical trials of cell-based therapy using stem cells such as mesenchymal stem cells (MSCs) have been conducted [46,47]. However, in most cases, the optimal therapeutic effect of transplanted cells has not been achieved due to low engraftment rates and short survival durations after transplantation [48,49,50]. Recently, many researchers have attempted to improve these therapeutic effects by functionalizing these cells. In addition, the biodistribution and fate of transplanted cells are unclear due to a lack of in vivo cell tracking methods. It is important to understand the biodistribution and biological fate of transplanted cells in order to develop a therapeutic strategy for cell-based therapies. Cell tracking techniques provide information about the in vivo behavior of transplanted cells, including migration, translocation, proliferation, cell death, and differentiation [51,52,53]. Researchers have also attempted to develop a cell tracking method. In this section, we summarize some applications of click chemistry for cell engineering in cell transplantation.

### 2.1. Click Chemistry as a Tracking Tool for Transplanted Cells

Non-invasive cell tracking methods are widely used to monitor transplanted cells in real time [50,54,55]. They are classified into two categories, defined by the use of either reporter proteins or imageable probes. The cell tracking method using reporter proteins, such as fluorescent or luminescent proteins, is suitable for the imaging of living cells [56,57,58]. In addition, this tracking method is beneficial for the long-term tracking of cells that stably express the proteins. However, this method requires genetic engineering before transplantation, which may lead to changes in cellular characteristics. On the other hand, the cell tracking method using imageable probes can label cells with a variety of probes, including fluorescence dyes, reporter proteins, and radiotracers. As such, the labeled cells can be monitored not only by fluorescence and bioluminescence imaging but also by positron emission tomography (PET), computed tomography (CT), and magnetic resonance (MR) imaging after transplantation [59,60,61,62]. MR imaging is able to detect a small number of transplanted cells in deep tissues [63]. Furthermore, the cell tracking method involves a short reaction time and label cells with high efficacy. However, the detectable duration of this method is limited because of the instability of the labels [51,64]. In addition, imageable probes such as lipophilic dyes and radiotracers often cause cytotoxicity [50,64]. Recently, a new cell labeling method with imageable probes has been developed involving the application of metabolic glycoengineering and copper-free click chemistry (Figure 2A). The combination of metabolic glycoengineering and copper-free click chemistry allows for the stable labeling of cells with various molecules without affecting the characteristics of the cells. Therefore, this method is expected to overcome the problems associated with the direct cell labeling method. The applications of copper-free click reactions for cell tracking are summarized in Table 3.

The first application of click chemistry to track transplanted cells was performed using Ac_4_ManNAz and DBCO-fluorophore [38]. Cells of human lung adenocarcinoma cell line A549 were incubated with Ac_4_ManNAz to generate azide groups on the surface of A549 cells and were transplanted to the liver of mice. DBCO-Cy5 was then administered into the mouse tail vein after transplantation of azide-labeled A549 cells. These experiments showed that A549 cells with DBCO-Cy5 were detected in the liver using ex vivo analysis. Although this method allows for the reduction of false signals derived from cell fragments that are phagocytosed by macrophages, the differences of in vivo distribution between azide-labeled cells and DBCO-Cy5 may affect the detectability or labeling efficacy. Then, Yoon et al. found that the combination of metabolic glycoengineering and SPAAC reaction was useful for detecting transplanted cells in vivo [29]. This method allows for the detection of transplanted cells over a longer time than a conventional lipophilic tracer without the impairment of cellular functions. They labeled chondrocytes with near-infrared (NIR) fluorescence dye by using Ac_4_ManNAz and dibenzylcyclooctyne-SETA 650 (DBCO-650), and subcutaneously transplanted DBCO-650-labeled chondrocytes into mice. Transplanted chondrocytes could be detected for 4 weeks using in vivo imaging analysis, and this labeling was found to have a minimal effect on the cartilage formation of chondrocytes. Lee et al. also demonstrated that metabolic glycoengineering and the SPAAC reaction was useful for tracking transplanted stem cells [31]. Since MSCs are able to home to inflammation sites and tumor tissues in response to chemokines released from inflammation sites or tumors [46,65,66], the migration of MSCs was traced. Lee et al. labeled adipose-derived mesenchymal stem cells (ASCs) with Cy5, a fluorescence dye, using Ac_4_ManNAz and DBCO-Cy5 and confirmed the fluorescence signal from Cy5-labeled ASCs for 14 days, both in vitro and in vivo.

In addition, they demonstrated that the migration potency of Cy5-labeled ASCs toward ischemic lesions was maintained. Furthermore, Layek et al. intravenously administered Cy5.5-labeled MSCs prepared by using the combination of metabolic glycoengineering and the SPAAC reaction into tumor-bearing mice and normal mice [33]. Although the MSC-derived fluorescence signal was hardly detected in normal mice, the signal was detected in the tumors of tumor-bearing mice. In addition, labeling with Ac_4_ManNAz and DBCO-fluorophore had a minimal effect on the differentiation ability of MSCs into adipocytes and osteocytes.

### 2.2. Click Chemistry as a Tool for Cell-Based Drug Delivery

Since MSCs are able to home to tumor tissues, as described above, they have been applied as a drug delivery vehicle in cancer therapy [67,68]. Researchers have attempted to use MSCs as anti-cancer vehicles by loading them with anti-cancer drug-containing nanoparticles via endocytosis [69,70,71]. Systemically injected drug-loaded MSCs are reported to migrate to tumor tissues and release drugs by drug efflux transporters or external stimulus. However, drug loading with MSCs via endocytosis requires a long preparation (over 24 h). In addition, the drug loading capacity of MSCs is limited because of the small occupation of endosomes in the cytoplasm [72]. To overcome these limitations, cell surface modification using metabolic glycoengineering and copper-free click chemistry could be used for the functionalization of MSCs with nanoparticles (Figure 3B).

In 2017, Lee et al. developed a simple cell surface modification method using a combination of metabolic glycoengineering and the SPAAC reaction [35]. First, they prepared BCN-modified imageable chitosan nanoparticles (BCN-CNPs). CNPs can easily encapsulate various drugs with high biocompatibility and are widely used for drug delivery [73]. They encapsulated Cy5.5, iron oxide nanoparticles, and gold nanoparticles into the BCN-CNPs for optical imaging, MR imaging, and CT imaging. Next, BCN-CNPs were added to azide-labeled MSCs prepared using Ac_4_ManNAz to obtain BCN-CNPs-Cy5.5-labeled MSCs. In this method, BCN-CNPs were modified to the cell surface of MSCs for less than 1 h. Although the fluorescence signal from BCN-CNPs-Cy5.5-labeled MSCs was detected for 5 days under in vitro culture conditions, the fluorescence signal gradually decreased with time. On the other hand, the fluorescence signals from BCN-CNPs-Cy5.5 labeled MSCs were detected for 15 days after subcutaneous transplantation into nude mice. In this paper, the authors demonstrated that the combination of metabolic glycoengineering and the SPAAC reaction allowed for the modification of MSCs with CNPs in a short reaction time, and that CNP-modified MSCs could be tracked over the long-term. On the other hand, these results indicate that the combination of metabolic glycoengineering and the SPAAC reaction may be a good tool to add anti-cancer functions to MSCs by modifying MSCs with anticancer agent-loaded CNPs. Therefore, the use of MSCs with CNPs that are stably loaded could be beneficial for drug delivery.

As such, metabolic glycoengineering can generate bioorthogonal chemical receptors on the surface of cells and therefore provides a binding site for nanoparticles. Copper-free click chemistry allows to modify cells with nanoparticles containing a variety of compounds in a short amount of time.

### 2.3. Applications of Click Chemistry for the Formation of Cell Complexes

The combination of metabolic glycoengineering and click chemistry is a powerful tool for cell surface modification. This cell surface modification method can be used for cell functionalization in cell-based sciences and engineering [23]. Koo et al. developed the cell gluing technique based on the iEDDA reaction (Figure 2C) [34]. Non-adherent human Jurkat T lymphocytes were treated with Ac_4_ManNAz followed by the addition of Tz-DBCO or TCO-DBCO. After incubation for 10 min, about 46% of the Tz- and TCO-modified Jurkat cells formed two- or multiple-cell pairs. The chemically glued cells were able to withstand a shear stress of over 20 dyn/cm^2^, which was higher than the typical vessel-wall shear stress levels of 1–6 dyn/cm^2^ in veins and 15 dyn/cm^2^ in arteries. Furthermore, after the intravenous injection of the glued cells, the cells survived and maintained their structures of two- or multiple-cell pairs in the bloodstream. The results showed that iEDDA reactions could be used to successfully carry out stable cell-cell adhesion in a short amount of time. Although there are few studies on the application of iEDDA reactions for cell engineering, the use of the iEDDA reaction has many benefits in cell engineering due to its fast reaction rate.

## 3. Applications of Click Chemistry for Drug Delivery in the Diagnosis and Treatment of Diseases

Recently, copper-free click chemistry has been applied for drug delivery in the diagnosis and treatment of diseases as an effective tissue-targeting method due to its high specificity, quick reaction rate, and stability. Some researchers have demonstrated that the labeling method using metabolic glycoengineering and copper-free click chemistry is useful for the visualization of glycans in vivo [26,74]. These studies showed that copper-free click chemistry could be a useful application tool for in vivo molecule imaging, cell tracking, and tissue engineering. Furthermore, copper-free click chemistry has been used for tissue-targeted delivery of imaging agents and anti-cancer agents. In cancer therapy, tumor tissue or cancer cell-specific drug delivery is highly desirable in order to improve therapeutic efficacy and avoid adverse effects [75]. Furthermore, cancer-specific drug delivery is needed for detecting cancer cells in order to determine tumor size and metastasis [76]. Although many tumor tissue- or cancer cell-specific delivery systems have been developed, cancer-specific drug delivery is yet to be fully achieved. This is due to the fact that current drug delivery systems do not show sufficient specificity to cancer and have non-specific biodistributions. Antibody-based drug delivery systems using monoclonal antibodies (mAb) binding to cancer-specific antigens are one of the most suitable and successful strategies for cancer-targeting therapy. For example, mAb-drug conjugates (ADCs) and mAb-coated nanoparticles successfully contribute to improving accumulation and cellular uptake in cancer cells [77,78] through the enhanced permeability and retention (EPR) effect [79,80]. However, antibody-based drug delivery systems have some limitations, including their heterogeneity and the downregulation of antigens in cancer cells due to long-term chemotherapy or chronic drug exposure [81]. Metabolic glycoengineering can artificially generate bioorthogonal chemical receptors, such as azide, on the surface of various types of cells, including cancer cells [20,24,40]. If bioorthogonal chemical receptors can be selectively introduced onto the surface of cancer cells, click chemistry with metabolic glycoengineering could overcome these limitations. Recently, researchers have attempted to develop a strategy which enables cancer cell-specific labeling with substrates for click chemistry, in an attempt to apply click chemistry and metabolic glycoengineering to cancer therapy and diagnosis. The application of metabolic glycoengineering and copper-free click chemistry for tumor targeting delivery and imaging are summarized in Table 4. In this section, we describe recent applications of click chemistry for drug delivery in the diagnosis and treatment of diseases.

### 3.1. Tumor-Specific Labeling with Azide Groups

The combination of metabolic glycoengineering and click chemistry is a powerful tool for the labeling and targeting of cancer cells [24,40]. Ac_4_ManNAz is widely used for the introduction of azide groups onto the surface of cancer cells because it can easily introduce azide groups both in vitro and in vivo without any apparent toxicity [28,74,82]. However, if systemically injected, Ac_4_ManNAz is distributed into many tissues, such that azide groups are introduced not only into the targeted tumor tissue but also into normal tissue [39,83]. Since non-specific azide labeling can have undesirable effects, it is important to selectively introduce azide groups into target cells for successful cancer-specific drug delivery, which can be achieved by using metabolic glycoengineering and click chemistry.

Koo et al. carried out successful tumor-targeting drug delivery using bioorthogonal chemical receptors and the SPAAC reaction [40]. They introduced azide groups into tumor cells in subcutaneous tumor-bearing mice by intratumorally injecting Ac_4_ManNAz. The amount of azide groups in the tumor tissues was found to increase in a dose-dependent manner. Furthermore, the accumulation of DBCO-modified liposomes in azide-expressing tumor tissue increased significantly after intravenous administration compared with untreated tumor tissues.

This result shows that azide-expressing tumor cells can be targeted by DBCO-modified drugs via the SPAAC reaction. However, the intratumoral injection of Ac_4_ManNAz poses a disadvantage since it has limitations in clinical application [103]. To specifically introduce azide groups into cancer cells, Ac_4_ManNAz analogs have been developed (Figure 3A). 

Chang et al. demonstrated that the cancer cell-specific azide labeling method using a cancer overexpressing enzyme-cleavable Ac_3_ManNAz analog and the SPAAC reaction was useful for drug delivery for cancer cell-specific imaging [104]. They synthesized an Ac_3_ManNAz conjugated to a substrate of the prostate-specific antigen (PSA), a serine protease that was highly secreted by prostate cancer [105]. This Ac_3_ManNAz analog is metabolized to 1,3,4-tri-*O*-acetyl-*N*-azidoacetylmannosamine (Ac_3_ManNAz) by PSA protease, and the azide groups are introduced into the glycans of the target cells after cellular uptake of Ac_3_ManNAz. They succeeded in effectively introducing azide groups into PSA-positive prostate cancer cell line PC-3 cells by using this Ac_3_ManNAz analog, and azide-labeled cells were imaged using DIFO-biotin and avidin-fluorophore. Shim et al. also successfully applied a cancer cell-specific azide labeling method using a cathepsin B-cleavable Ac_3_ManNAz analog (RR-S-Ac_3_ManNAz) [84]. First, they conjugated Ac_3_ManNAz with a cathepsin B-cleavable peptide (Lys-Gly-Arg-Arg, KGRR) using a self-immolative linker *p*-aminobenzyloxycarbonyl (*S*). The authors hypothesized that the designed RR-S-Ac_3_ManNAz would be specifically degraded by cathepsin B in the cytoplasm of the target cells and changed to S-Ac_3_ManNAz, which would produce Ac_3_ManNAz after hydrolysis. Cathepsin B is a cysteine protease overexpressed in various human tumors [106]. By the addition of RR-S-Ac_3_ManNAz, azide groups were specifically introduced into the surface of cathepsin B positive cancer cell lines, including human colon adenocarcinoma cell line HT29 cells, human breast cancer cell line MDA-MB-231 cells, PC-3 cells, and human glioma cell line U-87 MG cells. Moreover, the intravenous injection of RR-S-Ac_3_ManNAz introduced azide groups into cathepsin B-overexpressing tumor-bearing mice, and the tumor tissue was detected in the fluorescence imaging by intravenously injecting DBCO-Cy5.5. This study showed that a tumor targeting strategy based on metabolic glycoengineering and the SPAAC reaction was successful via intravenous injection. Wang et al. successfully applied a cancer cell-specific azide labeling method using a cancer-over expressing enzyme-cleavable Ac_3_ManNAz analog and the SPAAC reaction [41]. They synthesized an Ac_3_ManNAz analog which could be readily degraded by cancer-overexpressing protease, histone deacetylase and cathepsin L (DCL-AAM). Intravenous injection of DCL-AAM to tumor-bearing mice introduced azide groups into the tumor tissue at the same level as with tumor-bearing mice after intravenous injection of Ac_4_ManNAz at the same dose. On the other hand, the azide expression levels in normal tissues of DCL-AAM-treated mice were lower than those in Ac_4_ManNAz-treated mice. These results showed that the cancer cell-specific cleavable Ac_3_ManNAz analog reduced the expressing chemical receptors in normal tissues and improved the targeting efficacy of drugs via click chemistry.

As described above, Ac_4_ManNAz and its analogs are distributed not only to targeted tumor tissues but also to normal tissues after systemic injection. To achieve tumor-specific labeling with azide groups, researchers have attempted to control the pharmacokinetics of Ac_4_ManNAz. Here, we describe recent reports regarding the tumor tissue-specific delivery of Ac_4_ManNAz using nanomaterials (Figure 3B). Lee et al. synthesized Ac_4_ManNAz-loaded CNPs to improve the accumulation of Ac_4_ManNAz in tumor tissues [83]. The intravenous injection of Ac_4_ManNAz-loaded NPs to tumor-bearing mice specifically introduced azide groups into tumor tissues. Furthermore, the expression levels of azide groups in tumor tissues after injection of Ac_4_ManNAz-loaded CNPs were about 2-fold higher than those of Ac_4_ManNAz solution-treated groups. This is because most nanoparticles, including Ac_4_ManNAz-loaded CNPs, accumulate easily in tumor tissues as a result of the EPR effect. This study showed that intravenous injection of Ac_4_ManNAz-CNPs effectively and specifically generated azide groups in tumor tissues. Xie et al. also demonstrated that azide sugar-loaded folate-modified liposomes were useful for labeling cell-surface glycans of folate receptor-overexpressing cancer cells with azide groups [107]. Folate receptors are overexpressed in various cancer cells and have been widely used as target molecules for drug delivery in cancer therapy [108]. In this study, azide sugar-loaded folate modified-liposomes were effectively taken up by folate receptor-overexpressing HeLa cells via endocytosis, and could deliver sufficient amounts of azide sugars into the cytosol to introduce azide groups onto the surface of HeLa cells. Du et al. also demonstrated that Ac_4_ManNAz-loaded lipid nanomicelles generated azide groups into tumor tissues [43]. Moreover, Lee et al. developed nano-sized metabolic precursors (Nano-MPs) with a combined structure of generation 4 poly(amidoamine) dendrimer back bone and triacetylated *N*-azidoacetyl-d-mannosamine. The intravenous injection of Nano-MPs in tumor-bearing mice showed the localization of azide groups in tumor tissues, indicating that Ac_4_ManNAz was specifically delivered to tumor tissues within which azide groups were successfully generated [37]. As such, nano-sized carriers loaded or conjugated with Ac_4_ManNAz have been demonstrated to introduce sufficient amounts of azide groups into tumor cells to improve the accumulation of click chemistry chemical-drug conjugates. This method could be useful to specifically generate bioorthogonal chemical receptors in tumor tissue.

### 3.2. Tumor-Targeting Delivery by Click Chemistry for Cancer Therapy

To carry out cancer-targeting therapy safely and efficiently, cancer-targeting strategies using click chemistry chemical-drug conjugates and bioorthogonal chemical receptors have been reported. Wang et al. developed a tumor-targeting strategy using metabolic glycoengineering and DBCO-drug conjugate to enhance the accumulation of drugs in tumors (Figure 3C) [41]. First, they synthesized the DBCO-valine-cysteine-doxorubicin conjugate (DBCO-VC-Dox), which was degraded by a cancer-overexpressing cathepsin B protease followed by the release of Dox. DCL-AAM was intravenously injected into tumor-bearing mice daily for 3 days for the introduction of azide groups into the tumor tissue. Then, 24 h after the third injection, DBCO-VC-Dox was intravenously injected into DCL-AAM- or PBS-treated mice. The amount of DBCO-VC-Dox in the DCL-AAM-treated group was higher than that of the PBS group in the tumor tissue, while no significant changes in terms of DBCO-VC-Dox accumulation were observed in the normal tissue. Moreover, the combination of DCL-AAM and DBCO-VC-Dox significantly suppressed tumor growth in mice. This study showed that specific-tumor labeling with azide groups enhanced drug accumulation in targeted tumors and the therapeutic effect of anti-cancer agents via the SPAAC reaction.

Researchers have recently reported that modifying nanoparticles with click chemistry chemicals such as BCN and DBCO improves the affinity between nanoparticles and azide-labeled cancer cells. Lee et al. demonstrated that tumor targeting using two types of nanoparticles improved the accumulation of drug-containing nanoparticles in tumors [37]. First, they synthesized CNPs conjugated with BCN and chlorine e6 (BCN-Ce6-CNPs). Ce6, a photosensitizer, is highly activated by irradiation with NIR light and generates singlet oxygen [109,110]. Next, they intravenously injected Ac_4_ManNAz-loaded CNPs into tumor-bearing mice to introduce azide groups into the tumor tissue, and then irradiated the subcutaneous tumor tissue in mice with an NIR laser after the intravenous injection of BCN-Ce6-CNPs. After laser irradiation, a black scab was formed on the skin at the irradiation site in Ac_4_ManNAz-CNPs- and BCN-Ce6-CNPs-treated mice, due to the generation of excessive single oxygen in the tumor tissue. The tumor growth of Ac_4_ManNAz-CNPs- and BCN-Ce6-CNPs-treated mice was significantly inhibited. Layek et al. demonstrated that the MSCs-based targeting strategy improved the tumor targeting efficacy of drug-loaded nanoparticles [33]. They treated MSCs with Ac_4_ManNAz to label azide groups and found that azide labeling hardly affected the migration or viability of MSCs. Then, azide-labeled MSCs were intraperitoneally injected into metastatic ovarian tumor-bearing mice, followed by the intraperitoneal injection of paclitaxel-loaded DBCO-modified poly (lactide-co-glycolide) (PLGA) nanoparticles (DBCO-NPs). The administration of azide-labeled MSCs and DBCO-NPs showed significant inhibition of tumor growth and improved the overall survival of mice. These studies have shown that a two-step drug delivery strategy based on generating bioorthogonal chemical receptors and the SPAAC reaction could be useful as a cancer-targeting therapy.

### 3.3. Application of iEDDA Reaction for Tumor-Specific Delivery

Since the iEDDA reaction has the fastest reaction rate of the bioorthogonal reactions [23], its application in cancer-specific therapy and diagnosis is expected to achieve a more effective targeting and therapeutic efficacy than the SPAAC reaction. In this section, we discuss the application of the iEDDA reaction for cancer therapy and diagnosis. Rossin et al. were the first to apply the iEDDA reaction for cancer diagnosis (Figure 3D) [88]. They synthesized the CC49 (anti-tumor associated glycoprotein 72 monoclonal anti body [TAG72])-TCO conjugate (CC49-TCO) and intravenously injected it into colon cancer xenografts-bearing mice in an attempt to label tumor tissues with TCO, followed by the injection of DOTA-Tz labeled with ^111^In. The TCO-tagged tumors in the mice were then visualized using single-photon emission computed tomography/computed tomography (SPECT/CT) imaging. Zeglis et al. demonstrated that pretargeted PET imaging of the iEDDA reaction could be used to image tumor tissue with high quality [89]. They synthesized the TCO-modified humanized anti-A33 monoclonal antibody (A33-TCO) and intravenously injected it into subcutaneous A33 antigen-expressing xenografts-bearing mice. Then, ^64^Cu-NOTA-labeled Tz were intravenously injected into TCO-treated mice for PET imaging. ^64^Cu-NOTA-labeled Tz rapidly distributed and accumulated in the tumor tissue after injection, and a high tumor-to-background contrast was obtained at early time points. Although the mAb-radionuclide conjugate is conventionally used for cancer cell-specific PET and SPECT imaging, its long biological half-life requires high radiation doses [89,97]. These pretargeted methods based on the iEDDA reaction reduce the radiation dose to nontarget tissues. Furthermore, the efficiency of a pretargeting system based on the iEDDA reaction for cancer diagnosis by PET imaging has been demonstrated. Keinänen et al. showed a two-step tumor imaging method based on the iEDDA reaction [90]. They synthesized TCO-modified cetuximab (anti-epidermal growth factor receptor [EGFR] monoclonal antibody) or trastuzumab (anti-human epidermal growth factor receptor 2 [HER-2] monoclonal antibody; TCO-mAb) and then intravenously injected it into EGFR-positive or HER-2-positive tumor-bearing mice to label cancer cells with TCO, followed by the injection of ^18^F-radiolabeled Tz tracer ([^18^F] TAF). The iEDDA reaction has been used for the visualization of both antibodies and tumor tissues by PET imaging. Rossin et al. applied the iEDDA reaction for the selective cleavage of tumor-bound ADCs for cancer therapy [101]. First, they synthesized a CC49 conjugate with doxorubicin (DOX) via TCO linker (CC49-TCO-DOX), which was rapidly degraded by the iEDDA reaction. Next, CC49-TCO-DOX was intravenously injected into TAG72-overexpressing colon carcinoma xenografts-bearing mice, followed by the administration of Tz analog. After administration of Tz analog, CC49-TCO-DOX was cleaved and DOX was released into the tumor tissues. These findings demonstrate that the iEDDA reaction has the potential to improve the targeting efficacy of cancer therapy for its treatment and diagnosis.

## 4. Conclusions

In this review, we discussed the application of copper-free click chemistry for cell engineering in cell transplantation and drug delivery. Although certain factors remain unclear, including the safety of utilizing copper-free click chemistry and metabolic glycoengineering, these reports suggest that copper-free click chemistry is a powerful tool for cell engineering and drug delivery. As such, high biocompatible and fast copper-free click chemistry and its application in the biological and medical fields can be expected to develop further in the near future.

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
