# Peer review of "Click Chemistry as a Tool for Cell Engineering and Drug Delivery"

_molecules, 2019, doi:10.3390/molecules24010172_

Round 1
Reviewer 1 Report
Review comment to Molecules-408649
Authors present the application of SPAAC reaction in the cell engineering and drug delivery. This reviews cover interesting topic which is important for both fundamental biochemical investigation and application in biomedical field. However, this manuscript is not easy to understand for readers.
Here are my suggestions
1. Most reactions/preparation presented in this manuscript cover SPAAC reaction. Why authors still use the Click rather than “ SPAAC reaction” which his more precisely
2. There is no any figure or scheme to discuss the detail of chemical structure and biological experiment working mechanism. Strongly suggest authors should add reaction scheme in each section and the schematic presentation of the experiments related to biological study.
3. There are many abbreviation in this manuscript. A table to summarize all the used abbreviations are helpful.
4. There is not authors’ comment in this manuscript.
5. Author should consider adding one more section about the toxicity of the presented compounds which is important for biomedical application.
I cannot recommend to accept this manuscript with current status. Please revised and resubmit.
Reviewer 2 Report
Comments to the Authors
The review article titled "Click Chemistry as a tool for cell engineering and drug delivery" by Kusamori et al. summarize the importance of click chemistry for cell engineering and drug delivery. The review article contains large amount of detailed information, its structure and flow is good.
However it would be nice if the authors include some graphic images showing the importance of click chemistry in drug delivery and cell transplantation.
I recommend this review for publication in Molecules.
Reviewer 3 Report
This review work done by Takayama et al. summarized most recent progress of using click chemistry for cell engineering and drug delivery including cancer diagnosis and therapy. Using copper-free click chemistry for various in vivo applications is an interesting topic. And I believe this work will gain a wide range of interest from the readers of this journal and beyond. This review work is well organized and clearly presented. I would not hesitate to recommend this work to the editor.
Round 2
Reviewer 1 Report
I suggest accepting this manuscript with current format